# Isolation of Scalimides A–L: *β*-Alanine-Bearing Scalarane Analogs from the Marine Sponge *Spongia* sp.

**DOI:** 10.3390/md20110726

**Published:** 2022-11-18

**Authors:** A-Young Shin, Hyi-Seung Lee, Jihoon Lee

**Affiliations:** 1Korea Institute of Ocean Science & Technology (KIOST), Busan 49111, Republic of Korea; 2Department of Marine Biotechnology, University of Science & Technology, Daejeon 34113, Republic of Korea

**Keywords:** scalarane sesterterpene, scalimide, *Spongia* sp., marine sponge, antimicrobial activity

## Abstract

A chemical investigation of a methanol extract of *Spongia* sp., a marine sponge collected from the Philippines, identified 12 unreported scalarane-type alkaloids—scalimides A–L (**1**–**12**)—together with two previously described scalarin derivatives. The elucidation of the structure of the scalaranes based on the interpretation of their NMR and HRMS data revealed that **1**–**12** featured a *β*-alanine-substituted E-ring but differed from each other through variations in their oxidation states and substitutions occurring at C16, C24, and C25. Evaluation of the antimicrobial activity of **1**–**12** against several Gram-positive and Gram-negative bacteria showed that **10** and **11** were active against *Micrococcus luteus* and *Bacillus subtilis,* respectively, with MIC values ranging from 4 to 16 μg/mL.

## 1. Introduction

Scalaranes are one of the most prevalent groups of sesterterpenes, and they are primarily found in marine invertebrates [1,2,3]. Since scalarin, the first reported natural scalarane-type marine product, was discovered in the marine sponge *Cacospongia scalaris* in 1972 [4], several hundreds of scalarane sesterterpenes have been identified from sponges belonging to the order *Dicytoceratida* [5]. This class of scalaranes is characterized as a *trans*-fused 6/6/6/6 carbocyclic ring system and, in many cases, contains an additional pentacyclic E ring [6]. The general structure of the E ring contains an oxygen atom in the form of a lactone or furan, but certain reports have shown that a nitrogen-rich environment can produce a pyrrole or lactam [7]. The first example of nitrogen-bearing scalarane was found in the case of molliorins that were reported from 1977 to 1979. A series of investigations on the marine sponge *Cacospongia mollior* resulted in consecutive isolations of five pyrrole-bearing scalaranes—molliorins A–E [8,9,10,11]. Plausible biosynthesis of pyrrole in molliorins that involved condensation reactions of scalaridial—which is a common intermediate—with the corresponding amines was proposed, and this could be proved by the synthesis of molliorins after the proposed condensation under acidic conditions.

Since the mid-2000s, scalaranes containing lactams or cyclic imides have been identified. Hyatelactam, which was isolated from the marine sponge *Hyatella intestinalis* in 2006, is the first example of a lactam-bearing scalarane [12]. Following this report, 17 additional scalaranes belonging to this class were isolated from the marine sponges *Hyatella* sp., *Hyrtios* sp., *Petrosaspongia* sp., and *Spongia* sp. [13,14,15,16,17]. In these cases, a maleimide or *α*,*β*-unsaturated *γ*-butyrolactam moiety appeared as a structural feature of the E ring, but only hyrtioscalarin A contained a linear amide terminus (Figure 1). Although the pharmacological activity of the scalarane alkaloids has not been thoroughly investigated, some interesting biological properties have been reported, such as the anticancer activities of hyatelactam against HT-29 (human colorectal adenocarcinoma cell line, GI_50_ = 8.1 μM) [7], hyrtioscalarin D against A549 (adenocarcinomic human alveolar basal epithelial cell, IC_50_ = 5.4 μM), and hyrtioscalarin G against A549 (IC_50_ = 6.9 μM) and PC3 (human prostate cancer cell, IC_50_ = 5.2 μM) [12], as well as the inhibitory effect of petrosaspongiolactam A on the binding of TDP-43 to bt-TAR-32 (IC_50_ = 0.6 μM) [9] and the antibacterial activity of the sodium form of hyrtioscalarin F against *Bacillus subtilis* (MIC = 6.25 μg/mL) and of hyrtioscalarin A against *Klebsiella pneumoniae* (MIC = 2 μg/mL) [8].

The marine sponge of the genus *Spongia*, which belongs to the order *Dicytoceratida*, is known as a rich source of scalarane sesterterpenes. To date, various scalarane sesterterpenes exhibiting cytotoxicity or antagonistic activity against FXR (farnesoid X receptor), such as scalarolide [18], isoscalarafurans A–B [19], scalalactams A–D [16], and 12-*O*-deacetyl-12-*epi*-19-deoxy-21-hydroxyscalarin [20], have been isolated from marine sponges [21,22]. Our preliminary biological evaluation of the methanol extract of *Spongia* sp. collected from the Bohol province in the Philippines revealed a mild antimicrobial activity, prompting an investigation of bioactive secondary metabolites of the sponge. In this study, we report the identification of 12 undescribed scalarane alkaloids—scalimides A–L (**1**–**12**)—that contain a *β*-alanine moiety and their antimicrobial activity against several strains of Gram-positive and Gram-negative bacteria (Figure 2).

## 2. Results

### 2.1. Elucidation of the Structure

Scalimide A (**1**) was isolated as an amorphous solid. Its molecular formula was determined to be C_30_H_43_NO_7_ by HRESIMS (*m/z* 552.2925 [M + Na]^+^, as calculated for C_30_H_43_NO_7_Na, 552.2932), corresponding to 10 degrees of unsaturation (DOUs). Inspection of the ^1^H and ^13^C NMR spectra (Table 1, Appendix A) in combination with the HSQC spectrum revealed the presence of four carbonyl carbons (*δ*_C_ 174.4, 172.1, 171.9, 170.4), one tetrasubstituted double bond (*δ*_C_ 150.0, 141.9), two sp^3^ oxymethines (*δ*_H_ 5.52/*δ*_C_ 76.3, *δ*_H_ 4.59/*δ*_C_ 65.6), three sp^3^ methines, nine sp^3^ methylenes, and six methyl singlets, including one acetyl group (*δ*_H_ 1.93/*δ*_C_ 21.0, *δ*_H_ 1.32/*δ*_C_ 21.5, *δ*_H_ 0.98/*δ*_C_ 17.4, *δ*_H_ 0.86/*δ*_C_ 33.7, *δ*_H_ 0.87/*δ*_C_ 16.4, *δ*_H_ 0.84/*δ*_C_ 21.7). The HMBC correlations from H_3_-19/20 (*δ*_H_ 0.84/0.87) to C-3 (*δ*_C_ 43.2)/C-4 (*δ*_C_ 34.2)/C-5 (*δ*_C_ 58.0), from H_3_-21 (*δ*_H_ 0.98) to C-8 (*δ*_C_ 38.6)/C-9 (*δ*_C_ 54.1)/C-14 (*δ*_C_ 51.3), from H_3_-22 (*δ*_H_ 0.87) to C-1 (*δ*_C_ 40.9)/C-5/C-9/C-10 (*δ*_C_ 38.1), and from H_3_-23 (*δ*_H_ 1.32) to C-12 (*δ*_C_ 76.3)/C-13 (*δ*_C_ 41.6)/C-14/C-18 (*δ*_C_ 150.0) identified the 6/6/6/6 fused cyclic system of the scalarane scaffold (Figure 3). The 12-acetoxy group was established by the HMBC correlations from H-12 (*δ*_H_ 5.52)/12-C*H*_3_CO (*δ*_H_ 1.93) to 12-CH_3_*C*O (*δ*_C_ 173.5). Another oxymethine was inferred at C-16, as was evident from the spin system for H-14 (*δ*_H_ 1.66)–H_2_-15 (*δ*_H_ 2.21/1.57)–H-16 (*δ*_H_ 4.59) in the ^1^H-^1^H COSY spectrum. The calculated DOU value and the preliminarily identified functional groups suggested that **1** contained an extra E ring in order to form a pentacyclic skeleton. The HMBC correlations from H_3_-23 to C-18 and from H-16 to C-17 (*δ*_C_ 141.9)/C-18/C-24 (*δ*_C_ 171.9) indicated the presence of a pyrrole-2,5-dione moiety that was found in scalalactam A [16]. The presence of the propionic acid side chain was corroborated by the ^1^H-^1^H COSY cross-peak from H_2_-1′ (*δ*_H_ 3.69) to H_2_-2′ (*δ*_H_ 2.56), the HMBC correlations from H_2_-1′/H_2_-2′ to C-3′ (*δ*_C_ 174.4), and a strong IR absorption at 1735 cm^−1^. Additionally, the HMBC correlations from H_2_-1′ to C-24/C-25 (*δ*_C_ 170.4) suggested that the propionic acid moiety was connected to the pyrrole-2,5-dione moiety through the nitrogen atom.

The *trans*-fused cyclic system of **1** was determined by the consecutive NOESY correlations observed for H-3*α* (*δ*_H_ 1.15)–H-5 (*δ*_H_ 0.87)–H-9 (*δ*_H_ 1.23)–H-14 and H-6*β* (*δ*_H_ 1.48)–H_3_-22–H-11*β* (*δ*_H_ 1.78)–H_3_-21–H_3_-23 (Figure 4). The 12-acetoxy group was assigned as *α* based on the NOESY correlation from H-12 to H_3_-23, which was confirmed by the coupling constant between H-12 and H_2_-11 (dd, *J*_H-12-H-11_ = 3.5, 2.7 Hz). Similarly, the *β* configuration of 16-OH was assigned based on the NOESY correlation between H-14 and H-16, as well as the relatively large coupling constant between H-15*β* (*δ*_H_ 1.57) and H-16 (dd, *J*_H-16–H-15_ = 9.4, 7.0 Hz). Based on the biosynthetic origin, the absolute configurations of C-8, C-9, C-10, C-13, and C-14 in **1** were deduced to be identical to the reported scalarane alkaloids (Figure 1).

Scalimide B (**2**) was isolated as an amorphous solid. Its molecular formula was determined to be C_30_H_43_NO_7_ by HRESIMS (*m/z* 552.2931 [M + Na]^+^, as calculated for C_30_H_43_NO_7_Na, 552.2932), corresponding to 10 degrees of unsaturation. At first glance, the ^1^H NMR spectrum of **2** was almost identical to that of **1**, except for the smaller coupling constants observed for H-16 (*δ*_H_ 4.56, dd, *J* = 4.3, 1.5 Hz) and the appearance of H-15 (*δ*_H_ 1.93/1.83) at a lower frequency (Table 1), suggesting that **2** could be the 16-epimer of **1**. The complete structure could be proven by observing the HMBC correlations from H-16 to C-17 (*δ*_C_ 140.5)/C-18 (*δ*_C_ 151.1) and from H-14 (*δ*_H_ 2.03) to C-16 (*δ*_C_ 60.2). Additionally, the configuration of C-16 was further supported by the absence of the NOESY correlation between H-14 and H-16 and the higher chemical shift value of H-14 (*δ*_H_ 2.03 vs. *δ*_H_ 1.66 in **1**), which resulted from an increasing 1,3-diaxial interaction on the D ring due to an axial substitution of OH-16.

Scalimide C (**3**) was isolated as an amorphous solid. Its molecular formula was determined to be C_31_H_45_NO_7_ by HRESIMS (*m/z* 566.3087 [M + Na]^+^, as calculated for C_31_H_45_NO_7_Na, 566.3088), corresponding to 10 degrees of unsaturation. The ^1^H and ^13^C NMR spectra of **3** resembled those of **1**, except for the presence of a methoxy group at *δ*_H_ 3.52/*δ*_C_ 58.0, suggesting that **3** was a methyl ether of **1** at C-16, which was consistent with its being 14 mass units higher than that of **1** in the HRESIMS spectrum (Table 1). The location of the methoxy group was confirmed by the HMBC correlation from the methyl singlet at *δ*_H_ 3.52 to C-16 (*δ*_H_ 4.29/*δ*_C_ 74.9) (Figure 3). The configuration of C-16 was assigned as *β* based on the NOESY correlation between H-14 and H-16, as well as the large coupling constant for H-16 (*δ*_H_ 4.29, *J*_H-16–H-15_ = 9.1, 7.0 Hz).

Scalimide D (**4**) was isolated as an amorphous solid. The molecular formula of **4** was not only identical to that of **3**, but its ^1^H NMR spectrum also showed a high degree of similarity with that of **2**, except for the presence of a methoxy group at *δ*_H_ 3.46/*δ*_C_ 57.9, suggesting that **4** was the OMe-16 analog of **2** (Table 1, Appendix A). The HMBC correlation from the methyl singlet at *δ*_H_ 3.46 to C-16 (*δ*_C_ 70.1) confirmed the substitution of OMe at C-16, and its *α* orientation of the methoxy group was determined by the coupling constants of H-16 (*δ*_H_ 4.18, dd, *J*_H-16–H-15_ = 4.0, 1.6 Hz).

Scalimide E (**5**) was isolated as an amorphous solid, and its molecular mass was 14 mass units higher than those of **3** and **4,** as determined by HRESIMS (*m/z* 580.3240 [M + Na]^+^, as calculated for C_32_H_47_NO_7_Na, 580.3245). Compared with that of **4**, the ^1^H NMR spectrum of **5** showed an additional methyl singlet at *δ*_H_ 3.63, suggesting that **5** is a methyl ester of **4** (Table 1, Appendix A). The HMBC correlation from the methyl group at *δ*_H_ 3.63 to C-3′ (*δ*_C_ 173.0) confirmed the presence of a methyl ester moiety in the side chain of **5** (Figure 3). Scalimide E (**5**) could be an artifact generated from **4** during the course of purification using MeOH and TFA, but this possibility was not investigated in this study.

Scalimide F (**6**) was isolated as an amorphous solid. Its molecular formula was determined to be C_30_H_41_NO_6_ by HRESIMS (*m/z* 534.2817 [M + Na]^+^, as calculated for C_30_H_41_NO_6_Na, 534.2826), corresponding to 11 degrees of unsaturation. The analysis of the ^1^H and ^13^C NMR spectra in combination with the HSQC data, revealed the presence of a disubstituted double bond at *δ*_H_ 6.46/*δ*_C_ 138.3 and *δ*_H_ 6.39/*δ*_C_ 117.7 instead of the 16-OH in **1** (Table 1, Appendix A). The location of the double bond was determined to be Δ^15^ by the ^1^H-^1^H COSY cross-peaks from H-14 (*δ*_H_ 2.75) to H-15 (*δ*_H_ 6.46) and from H-15 to H-16 (*δ*_H_ 6.36), which was confirmed by the HMBC correlations from H-15 to C-14 (*δ*_C_ 54.9)/C-17 (*δ*_C_ 137.7) and from H-16 to C-15 (*δ*_C_ 138.3)/C-17/C-18 (*δ*_C_ 143.5).

Scalimide G (**7**) was isolated as an amorphous solid. Its molecular formula was determined to be C_31_H_47_NO_7_ by HRESIMS (*m/z* 568.3238 [M + Na]^+^, as calculated for C_31_H_47_NO_7_Na, 568.3245), corresponding to nine degrees of unsaturation. Analysis of the 1D and 2D NMR data revealed a similar pentacyclic framework to that of **4** (Table 2, Appendix A). However, one additional sp^3^ oxymethine at *δ*_H_ 5.39/*δ*_C_ 81.7 was observed as a substitution for one of the carbonyl groups of a pyrrole-2,5-dione in **4**. This information indicates the reduction of a carbonyl group in the pyrrole-2,5-dione of **4** to a hydroxyl group whose position was determined to be at C-24, as observed from the HMBC correlations from H-16 (δ_H_ 4.02)/H_2_-1′ (δ_H_ 3.68/3.53) to C-24 (δ_C_ 81.7) (Appendix A). Similarly, the small coupling constant of H-16 (dd, *J*_H-16-H-15_ = 4.4, 1.3 Hz) could be a sign of the *α*-orientation of OMe-16. Furthermore, the 24-OH configuration was assigned as a *β*-orientation based on the NOESY correlation between H-24 and 16*α*-OMe.

Scalimide H (**8**) was isolated as an amorphous solid. The ^1^H and ^13^C NMR data of **8** differed from those of **7** only in the presence of an additional methyl singlet at *δ*_H_ 3.04/*δ*_C_ 50.5, which was found to be in agreement with the HRESIMS result, indicating a molecular mass that was 14 units higher than that of **7** (Table 2, Appendix A). The HMBC correlations from *δ*_H_ 3.04 to C-24 (*δ*_C_ 87.0) and from H-24 (*δ*_H_ 5.44) to *δ*_C_ 50.5 could be evidence of the OMe-24 substituent (Figure 5). Since OMe-24 exhibited a cross-peak with H_3_-23 in the NOESY spectrum, the OMe-24 was determined to be *β*-oriented (Appendix A).

Scalimide I (**9**) was isolated as an amorphous solid. Its molecular formula was determined to be C_30_H_45_NO_6_ by HRESIMS (*m/z* 538.3129 [M + Na]^+^, as calculated for C_30_H_45_NO_6_Na, 538.3139), corresponding to nine degrees of unsaturation. The ^1^H and ^13^C NMR data revealed that **9** had the same scaffold as **1**, but with one methylene at *δ*_H_ 4.07/3.97 instead of a carbonyl group (Table 2, Appendix A). The HMBC correlations from *δ*_H_ 4.07/3.97 to C-17 (*δ*_C_ 153.9)/C-18 (*δ*_C_ 140.1)/C-25 (*δ*_C_ 171.7) indicated a *γ*-lactam moiety, and the NOESY correlation between one of the methylene protons at *δ*_H_ 3.97 and H-16 suggested the location of the methylene at C-24 to determine a 24-2*H*-pyrrol-25-one (Figure 5 and Appendix A). In addition, the *β*-configuration of OH-16 was confirmed by the large coupling constant of H-16 (dd, *J*_H-16–H-15_ = 9.8, 6.6 Hz).

Scalimide J (**10**) was isolated as an amorphous solid. The HRESIMS analysis of **10** revealed an identical molecular formula to that of **9** (*m/z* 516.3317 [M + H]^+^, as calculated for C_30_H_46_NO_6_, 516.3320). The ^1^H NMR spectrum of **10** featured a methylene at *δ*_H_ 4.06/3.77 (Table 2, Appendix A), which exhibited HMBC correlations to C-17 (*δ*_C_ 133.4)/C-18 (*δ*_C_ 162.3), confirming the presence of the *γ*-lactam moiety (Figure 5). However, the inversion of the chemical shift between C-17 and C-18 (*δ*_C-18_ > *δ*_C-17_), in comparison with that of **9**, suggested a 25-2*H*-pyrrol-24-one moiety, which was further evidenced by the NOESY correlations from H_2_-25 (*δ*_H_ 4.06/3.77) to H_3_-23 (*δ*_H_ 1.26). The *β*-orientation of OH-16 was determined by observing the consecutive NOESY correlations for H-5–H-9–H-14–H-16 (Figure 6).

Scalimide K (**11**) was isolated as an amorphous solid. Its molecular formula was determined to be C_30_H_43_NO_5_ by HRESIMS (*m/z* 520.3030 [M + Na]^+^, as calculated for C_30_H_43_NO_5_Na, 520.3033), corresponding to 10 degrees of unsaturation. The ^1^H NMR spectra of **11** in combination with the HSQC data showed olefinic protons at *δ*_H_ 6.25/6.03 and a methylene proton at *δ*_H_ 4.06 (Table 2, Appendix A). The double bond between C-15 and C-16 was substantiated by the ^1^H-^1^H COSY correlations from H-14 (*δ*_H_ 2.68) to H-15 (*δ*_H_ 6.03) and from H-15 to H-16 (*δ*_H_ 6.25). Similarly to **10**, the NOESY correlation between the methylene protons at *δ*_H_ 4.06 and H_3_-23 (*δ*_H_ 1.06) suggested the presence of a 25-2*H*-pyrrol-24-one moiety in **11**.

Scalimide L (**12**) was isolated as an amorphous solid. The ^1^H NMR spectrum of **12** was almost identical to that of **11**, except for an additional methyl singlet at *δ*_H_ 3.65, suggesting that **12** was a methyl ester of **11** (Table 2, Appendix A). The presence of an additional methyl group was consistent with the molecular formula of C_31_H_45_NO_5_, as determined in an HRESIMS analysis (*m/z* 512.3366 [M + H]^+^, as calculated for C_31_H_46_NO_5_, 512.3371). Therefore, the HMBC correlation from *δ*_H_ 3.65 to C-3′ (*δ*_C_ 174.4) confirmed the presence of the methyl propionate side chain, which may have been an artifact generated during the separation steps when using MeOH as an eluent.

### 2.2. Biological Activities

As the MeOH extract of *Spongia* sp. exhibited mild antimicrobial activity in our preliminary screening, the minimal inhibitory concentrations (MICs) of scalimides A–L (**1**–**12**) were evaluated by using several strains of Gram-positive and Gram-negative bacteria (Table 3). Interestingly, **1**–**12** were more potent against Gram-positive bacteria but were mostly inactive against Gram-negative bacteria. Among these compounds, **10** displayed the broadest spectrum of inhibitory effects, especially against *Micrococcus luteus* and *Bacillus subtilis*, with MIC values of 8 and 4 μg/mL, respectively. Although most of the isolated compounds displayed weak to moderate antibacterial activity, the structural diversity of the scalimides—arising from different substitutions on C-16, C-24, and C-25—provided useful information regarding the structure–activity relationship (SAR): (1) Regarding the MIC values of **1**–**4**, the methyl ether at C-16 appeared to be more potent than the hydroxyl group for the activity toward *B. subtilis*; (2) when comparing **4** with **5** and **11** with **12**, the carboxylic acid moiety at C-3′ was identified as a crucial factor for the antimicrobial activity; (3) reduction of the carbonyl group at C-24 to a hydroxyl group (**4** to **7**) and a methylene group (**1** to **9**) decreased antibacterial activity; (4) reduction of the carboxyl group of the imide at C-25 to a lactam (**1** vs. **10** and **6** vs. **11**) increased antibacterial activity at least four-fold. Additionally, the cytotoxicity of **1**–**12** against MCF7 (a breast cancer cell line) was evaluated to identify it as a potent anticancer agent, but only moderate anticancer effects were observed for all of the tested compounds.

## 3. Materials and Methods

### 3.1. General Experimental Procedures

Specific optical rotations were measured using a Rudolph Research Analytical (Autopol III) polarimeter (Rudolph Research Analytical, Hackettstown, NJ, USA). IR spectra were recorded using a JASCO FT/IR-4100 spectrophotometer (JASCO Corporation, Tokyo, Japan). The 1D (^1^H and ^13^C) and 2D (COSY, HSQC, HMBC, and NOESY) NMR spectra were recorded in CD_3_OD using a Bruker 600 MHz spectrometer (Bruker BioSpin GmbH, Rheinstetten, Germany) at 297.1 K. The ^1^H NMR spectra were collected from 32–64 scans, and the ^13^C NMR spectra were collected from 5000–15,000 scans, depending on the sample concentrations. The mixing time for the NOESY experiments was set to 0.33 s. Chemical shifts were reported in parts per million relative to CD_3_OD (*δ*_H_ 3.31, *δ*_C_ 49.0). High-resolution mass spectra were obtained using a Sciex X500R Q-TOF spectrometer (Framingham, MA, USA) equipped with an electrospray ionization (ESI) source. MPLC (medium-pressure liquid chromatography) was performed using a TELEDYNE ISCO CombiFlash Companion with a YMC-Dispopack AT ODS-25 40 g column (YMC Co. Ltd., Kyoto, Japan). HPLC (high-performance liquid chromatography) was performed using a PrimeLine Binary pump (Analytical Scientific Instruments, Inc., El Sobrante, CA, USA) equipped with a Shodex RI-101 (Shoko Scientific Co. Ltd., Yokohama, Japan) and UV-M201 using C18 columns (YMC-Triart C18, 250 × 10 mm I.D., or 250 × 4.6 mm I.D., 5 µm; YMC Co. Ltd., Kyoto, Japan).

### 3.2. Biological Material

Biological material was collected in March 2015 in the Philippines (9°45′32.31″ N 124°35′53.60″ E) at a depth of 15 m by scuba diving. The sponges were frozen at −20 °C until they were identified as *Spongia* sp. and chemically analyzed. A voucher sample (153PIL-209) was stored at the Marine Biotechnology Research Center, Korea Institute of Ocean Science and Technology (KIOST).

### 3.3. Extraction and Isolation

The specimens (wet wt.: 1.5 kg) were lyophilized and extracted repeatedly with MeOH (2.0 L × 2) and CH_2_Cl_2_ (2.0 L × 2) at room temperature. The extracts were combined and concentrated under reduced pressure. The residue (78.3 g) was partitioned with *n*-butanol (3.0 L) and water (3.0 L) to yield 35.5 g of an organic-soluble material. The *n*-butanol layer was further partitioned into *n*-hexane (2.0 L) and 15% aqueous methanol (2.0 L). The 15% aqueous methanol fraction (30.1 g) was concentrated and subjected to flash column chromatography over C18 (YMC Gel ODS-A, 60 Å, 230 mesh (YMC Co., Ltd., Kyoto, Japan)) with a stepwise gradient solvent system (50%, 60%, 70%, 80%, 90% *aqueous*-MeOH, and 100% MeOH, acetone, and EtOAc).

The 90% MeOH fraction (780.6 mg) was further separated using MPLC on a C18 column with a gradient solvent system from 70% *aq*-MeOH to 100% MeOH over 30 min to yield five fractions. The fourth fraction (566.9 mg) was separated using MPLC on a C18 column with a gradient solvent system from 60% MeOH to 100% MeOH over 40 min to yield four subfractions. The second subfraction (253.8 mg) was separated using HPLC (eluent 60% MeCN with 0.1% TFA) to yield **1** (3.5 mg, *t*_R_ = 60 min), **2** (3.8 mg, *t*_R_ = 62 min), **9** (1.4 mg, *t*_R_ = 30 min), and **10** (1.9 mg, *t*_R_ = 40 min). The third subfraction (197.9 mg) was separated using HPLC (eluent 70% MeCN with 0.1% TFA) to yield **3** (2.3 mg, *t*_R_ = 66 min), **4** (6.7 mg, *t*_R_ = 65 min), **5** (2.3 mg, *t*_R_ = 67 min), **7** (4.0 mg, *t*_R_ = 30 min), **8** (2.9 mg, *t*_R_ = 58 min), **11** (4.6 mg, *t*_R_ = 42 min), and **12** (1.1 mg, *t*_R_ = 43 min). The fourth subfraction (37.8 mg) was separated using HPLC (eluent 80% MeCN with 0.1% TFA) to yield **6** (3.2 mg, *t*_R_ = 58 min), scalarin (12.0 mg, *t*_R_ = 50 min; the structure is shown in the Supporting Information), and 12*α*-acetoxy-19*β*-hydroxyscalara-15,17-*dien*-20,19-olide (6.1 mg, *t*_R_ = 61 min).

**1**: Amorphous powder; ${\left[\alpha \right]}_{D}^{20}$ +90 (*c* 0.1, MeOH); IR (ATR) *ν*_max_ 3742, 2933, 2858, 1738, 1702, 1547, 1511, 1452, 1381, 1240, 1194 cm^−1^; ^1^H NMR and ^13^C NMR, see Table 1 and Appendix A; HRESIMS *m/z* 552.2925 [M + Na]^+^ (calculated for C_30_H_43_NO_7_Na, 552.2932).

**2**: Amorphous powder; ${\left[\alpha \right]}_{D}^{20}$ +100 (*c* 0.1, MeOH); IR (ATR) *ν*_max_ 3441, 2929, 2855, 1710, 1653, 1448, 1395, 1233, 1042 cm^−1^; ^1^H NMR and ^13^C NMR, see Table 1 and Appendix A; HRESIMS *m/z* 552.2931 [M + Na]^+^ (calculated for C_30_H_43_NO_7_Na, 552.2932).

**3**: Amorphous powder; ${\left[\alpha \right]}_{D}^{20}$ +86.7 (*c* 0.1, MeOH); IR (ATR) *ν*_max_ 2933, 2851, 1742, 1702, 1515, 1388, 1236, 1194 cm^−1^; ^1^H NMR and ^13^C NMR, see Table 1 and Appendix A; HRESIMS *m/z* 566.3087 [M + Na]^+^ (calculated for C_31_H_45_NO_7_Na, 566.3088).

**4**: Amorphous powder; ${\left[\alpha \right]}_{D}^{20}$ +93.3 (*c* 0.2, MeOH); IR (ATR) *ν*_max_ 2929, 1738, 1710, 1458, 1399, 1236, 1197, 1038 cm^−1^; ^1^H NMR and ^13^C NMR, see Table 1 and Appendix A; HRESIMS *m/z* 566.3083 [M + Na]^+^ (calculated for C_31_H_45_NO_7_Na, 566.3088).

**5**: Amorphous powder; ${\left[\alpha \right]}_{D}^{20}$ +43.3 (*c* 0.1, MeOH); IR (ATR) *ν*_max_ 2933, 2858, 1742, 1706, 1547, 1515, 1388, 1236, 1201 cm^−1^; ^1^H NMR and ^13^C NMR, see Table 1 and Appendix A; HRESIMS *m/z* 580.3240 [M + Na]^+^ (calculated for C_32_H_47_NO_7_Na, 580.3245).

**6**: Amorphous powder; ${\left[\alpha \right]}_{D}^{20}$ +86.6 (*c* 0.1, MeOH); IR (ATR) *ν*_max_ 2925, 1742, 1710, 1692, 1543, 1526, 1511, 1374, 1236, 1204 cm^−1^; ^1^H NMR and ^13^C NMR, see Table 1 and Appendix A; HRESIMS *m/z* 534.2817 [M + Na]^+^ (calculated for C_30_H_41_NO_6_Na, 534.2826).

**7**: Amorphous powder; ${\left[\alpha \right]}_{D}^{20}$ +56.6 (*c* 0.1, MeOH); IR (ATR) *ν*_max_ 3738, 2936, 2861, 1721, 1706, 1692, 1533, 1469, 1250 cm^−1^; ^1^H NMR and ^13^C NMR, see Table 2 and Appendix A; HRESIMS *m/z* 568.3238 [M + Na]^+^ (calculated for C_31_H_47_NO_7_Na, 568.3245).

**8**: Amorphous powder; ${\left[\alpha \right]}_{D}^{20}$ +53.3 (*c* 0.1, MeOH); IR (ATR) *ν*_max_ 3742, 2933, 2855, 1734, 1706, 1689, 1554, 1462, 1388, 1240 cm^−1^; ^1^H NMR and ^13^C NMR, see Table 2 and Appendix A; HRESIMS *m/z* 582.3395 [M + Na]^+^ (calculated for C_32_H_49_NO_7_Na, 582.3401).

**9**: Amorphous powder; ${\left[\alpha \right]}_{D}^{20}$ +60 (*c* 0.1, MeOH); IR (ATR) *ν*_max_ 3399, 2929, 2869, 1717, 1674, 1455, 1261, 1204, 1133 cm^−1^; ^1^H NMR and ^13^C NMR, see Table 2 and Appendix A; HRESIMS *m/z* 538.3129 [M + Na]^+^ (calculated for C_30_H_45_NO_6_Na, 538.3139).

**10**: Amorphous powder; ${\left[\alpha \right]}_{D}^{20}$ +113.3 (*c*, 0.1, MeOH); IR (ATR) *ν*_max_ 3420, 2925, 2855, 1717, 1674, 1466, 1374, 1246, 1194 cm^−1^; ^1^H NMR and ^13^C NMR, see Table 2 and Appendix A; HRESIMS *m/z* 516.3317 [M + H]^+^ (calculated for C_30_H_46_NO_6_, 516.3320).

**11**: Amorphous powder; ${\left[\alpha \right]}_{D}^{20}$ +183.3 (*c* 0.2, MeOH); IR (ATR) *ν*_max_ 2929, 2855, 1731, 1699, 1689, 1660, 1551, 1505, 1462, 1240 cm^−1^; ^1^H NMR and ^13^C NMR, see Table 2 and Appendix A; HRESIMS *m/z* 520.3030 [M + Na]^+^ (calculated for C_30_H_43_NO_5_Na, 520.3033).

**12**: Amorphous powder; ${\left[\alpha \right]}_{D}^{20}$ +56.6 (*c* 0.1, MeOH); IR (ATR) *ν*_max_ 2936, 1749, 1702, 1692, 1646, 1551, 1526, 1509 cm^−1^; ^1^H NMR and ^13^C NMR, see Table 2 and Appendix A; HRESIMS *m/z* 512.3366 [M + H]^+^ (calculated for C_31_H_46_NO_5_, 512.3371).

### 3.4. Assay

#### 3.4.1. Antimicrobial Assays

The test of minimal inhibitory concentrations (MICs) for **1**–**12** was carried out by using six bacterial strains: *Micrococcus luteus* (KCTC-1915), *Staphylococcus aureus* (KCTC-1927), *Bacillus subtilis* (KCTC-1021), *Escherichia coli* (KCTC-2441), *Salmonella typhimurium* (KCTC-2515), and *Klebsiella pneumoniae* subsp. (KCTC-2690). Bacteria were streaked onto Mueller–Hinton agar (MHA) plates and incubated for 24 h at 37 °C. A single selected colony was transferred to Muller–Hinton broth (MHB), cultured for 24 h at 37 °C, and harvested through centrifugation at 250 rpm. The turbidity of the bacterial suspensions was adjusted with the MacFarland standard 0.5 to 1.5 × 10^8^ cfu/mL by adding sterile MHB. Pure **1**–**12** was dissolved in DMSO and dispensed into 96-well plates at a concentration of 128 μg/mL. These compounds were serially diluted to 0.25 μg/mL. Subsequently, the bacterial suspensions were inoculated in 96-well plates and incubated for 15–20 h at 37 °C. The lowest concentrations that inhibited bacterial growth were recorded as the MIC values. Kanamycin (K1377) was purchased from the Sigma Chemical Company (St. Louis, MO, USA) and used as a positive control.

#### 3.4.2. Cytotoxicity Assays

An MTS assay was performed using the CellTiter 96^®^AQ_ueous_One Solution Cell Proliferation Assay (MTS) (Promega, Madison, WI, USA). MCF-7 cells were plated in 384-well plates at a density of 3700 cells/well. The seeded cells were incubated for 24 h at 37 °C under a 5% (*v*/*v*) CO_2_ atmosphere, were treated with **1**–**12** at seven different concentrations (1.25, 2.5, 5.0, 10, 20, 40, and 80 μΜ), and then incubated for another 48 h. DMSO was used as a vehicle control and staurosporine was used as a positive control. Viable cell numbers were determined by the concentration of formazan resulting from the tetrazolium conversion, and the absorbance was measured at 490 nm using an EnVision Xcite Multilabel Reader (PerkinElmer, Waltham, MA, USA).

## 4. Conclusions

In summary, the investigation of the antibacterial components of the marine sponge *Spongia* sp. resulted in the isolation of 12 unreported *β*-alanine-bearing scalaranes (**1**–**12**), which were named scalimides A–L. The structures of **1**–**12** were unambiguously determined by using conventional spectroscopic methods with 1D and 2D NMR data and HRESIMS. The structural differences in **1–12** arose from the variations in the oxidation states at C-24 and C-25, producing an imide, aminal, or lactam, as well as from substitutions at C-16 and C-24. Evaluation of MIC values against selected Gram-positive and Gram-negative bacteria identified **10** as the most potent antimicrobial agent among the isolated compounds, providing insights into the structure–activity relationships. Further studies aimed at identifying potent antimicrobial natural products from the sponge *Spongia* sp. are currently underway.

## Figures and Tables

**Figure 1 marinedrugs-20-00726-f001:**
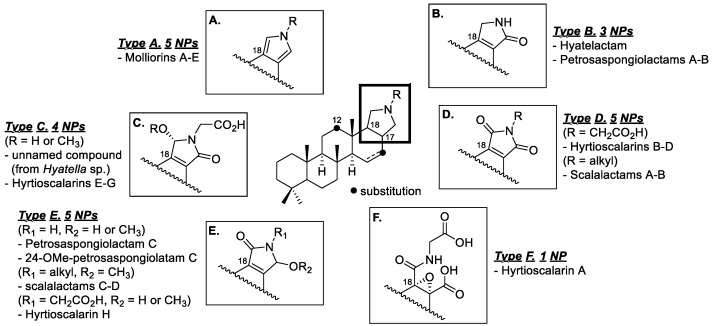
General structures of the E rings of scalarane alkaloids.

**Figure 2 marinedrugs-20-00726-f002:**
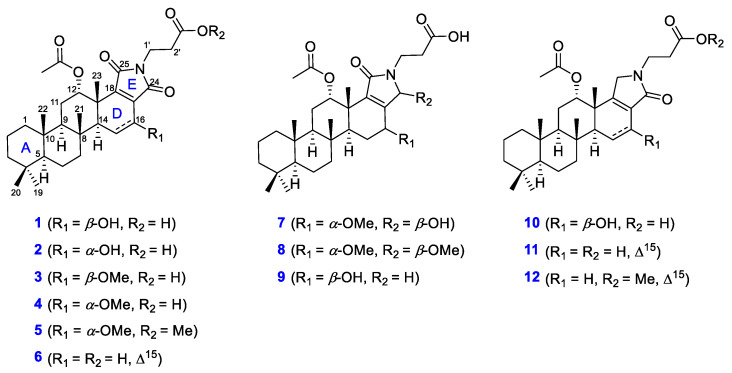
Structures of scalimides A–L (**1**–**12**) isolated from the marine sponge *Spongia* sp.

**Figure 3 marinedrugs-20-00726-f003:**
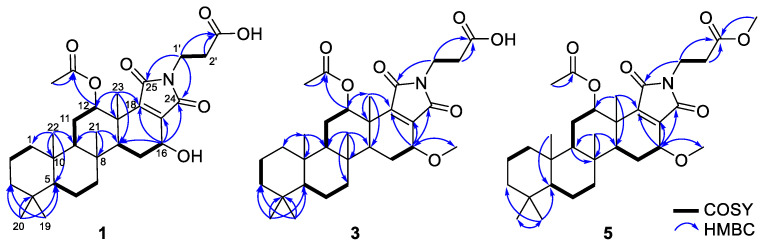
Key COSY and HMBC correlations observed for **1**, **3**, and **5**.

**Figure 4 marinedrugs-20-00726-f004:**
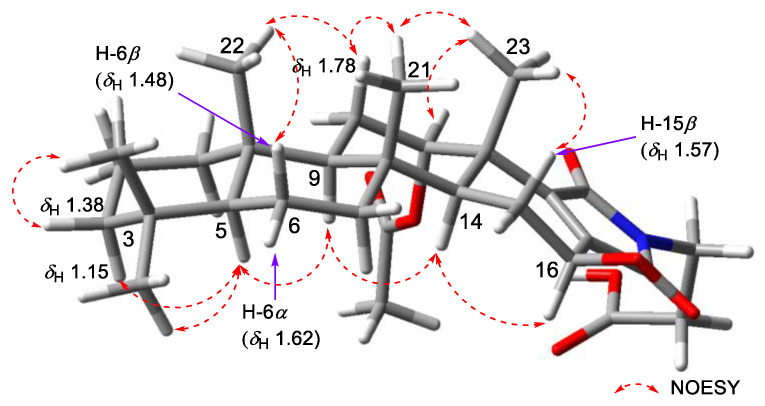
Key NOESY correlations of **1**.

**Figure 5 marinedrugs-20-00726-f005:**
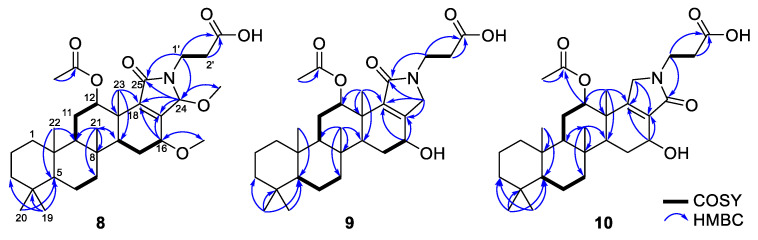
Key COSY and HMBC correlations observed for **8**, **9**, and **10**.

**Figure 6 marinedrugs-20-00726-f006:**
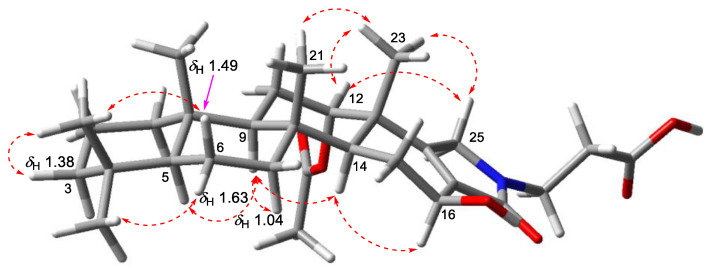
NOESY correlations of **10**.

**Table 1 marinedrugs-20-00726-t001:** The ^13^C (150 MHz) and ^1^H (600 MHz) NMR data for **1**−**6** ^a^.

Position	1		2		3		4		5		6	
	*δ*c	*δ*_H_ (*J* in Hz)	*δ*c	*δ*_H_ (*J* in Hz)	*δ*c	*δ*_H_ (*J* in Hz)	*δ*c	*δ*_H_ (*J* in Hz)	*δ*c	*δ*_H_ (*J* in Hz)	*δ*c	*δ*_H_ (*J* in Hz)
1	40.9	1.59, m 0.63, m	40.9	1.61, m 0.65, m	40.9	1.59, m 0.63, m	40.9	1.59, m 0.65, td (12.8, 3.7)	40.9	1.60, m 0.65, m	40.9	1.61, m 0.68, td (12.7, 3.8)
2	19.5	1.65, m 1.41, m	19.5	1.64, m 1.43, m	19.5	1.63, m 1.40, m	19.5	1.63, m 1.41, m	19.5	1.65, m 1.42, m	19.5	1.66, m 1.42, m
3	43.2	1.38, m 1.15, m	43.2	1.38, m 1.17, m	43.2	1.38, m 1.15, td (13.1, 3.9)	43.2	1.37, m 1.16, td (13.4, 4.0)	43.2	1.38, m 1.17, td (13.1, 3.9)	43.2	1.40, m 1.17, td (14.0, 13.5, 4.2)
4	34.2		34.2		34.2		34.6		34.2		34.3	
5	58.0	0.87, m	58.0	0.92, m	57.9	0.87, m	58.0	0.88, m	58.0	0.88, m	58.0	0.91, m
6	19.2	1.62, m 1.48, m	19.2	1.63, m 1.48, m	19.2	1.63, m 1.49, m	19.2	1.63, m 1.49, m	19.2	1.62, m 1.49, qd (13.2, 3.4)	18.9	1.65, m 1.53, m
7	42.4	1.88, m 1.03, m	42.3	1.84, m 1.11, m	42.3	1.89, dt (12.6, 3.5) 1.05, m	42.4	1.88, dt (12.6, 3.0) 1.03, m	42.4	1.88, dt (12.7, 3.5) 1.03, td (12.7, 3.9)	41.9	2.03, m 1.03, m
8	38.6		38.3		38.7		38.3		38.3		38.5	
9	54.1	1.23, m	54.3	1.32, m	54.1	1.22, m	54.3	1.25, dd (13.3, 2.3)	54.3	1.26, dd (13.4, 2.3)	53.4	1.31, m
10	38.1		38.1		38.1		38.1		38.1		38.0	
11	21.9	1.99, dd (12.6, 6.9) 1.78, ddd (15.2, 13.2, 2.4)	21.9	2.04, m 1.82, m	21.7	1.99, m 1.77, ddd (15.2, 13.2, 2.4)	21.9	2.01, dt (15.1, 3.0) 1.78, ddd (15.2, 13.3, 2.5)	21.9	2.01, dt (14.8, 2.8) 1.78, ddd (15.2, 13.2, 2.5)	22.1	2.06, m 1.76, ddd (15.2, 13.2, 2.4)
12	76.3	5.48, dd (3.5, 2.7)	76.3	5.52, t (2.9)	76.1	5.48, dd (3.2, 2.3)	76.2	5.49, dd (3.5, 2.3)	76.1	5.49, t (2.9)	74.7	5.44, dd (3.5, 2.3)
13	41.6		41.7		41.5		41.8		41.8		41.9	
14	51.3	1.66, m	46.6	2.03, m	51.0	1.65, m	46.9	1.92, m	46.9	1.94, dd (12.9, 1.6)	54.9	2.75, t (3.0)
15	28.6	2.21, dd (12.8, 6.9) 1.57, m	28.0	1.93, m 1.83, m	25.5	2.32, dd (12.8, 7.1) 1.59, m	23.5	2.11, dd (14.1, 1.6) 1.63, m	23.5	2.11, m 1.62, m	138.3	6.46, dd (9.7, 2.7)
16	65.6	4.59, dd (9.4, 7.0)	60.2	4.56, dd (4.3, 1.5)	74.9	4.29, dd (9.1, 7.0)	70.1	4.18, dd (4.0, 1.6)	70.0	4.12, dd (4.0, 1.6)	117.7	6.39, dd (9.7, 3.2)
17	141.9		140.5		140.6		139.1		139.2		137.7	
18	150.0		151.1		151.3		151.7		151.7		143.5	
19	33.7	0.87, s	33.8	0.87, s	33.7	0.87, s	33.7	0.87, s	33.8	0.87, s	33.8	0.88, s
20	21.7	0.84, s	21.7	0.85, s	21.7	0.85, s	21.8	0.85, s	21.7	0.85, s	21.8	0.85, s
21	17.4	0.98, s	17.6	0.96, s	17.4	0.98, s	17.6	0.97, s	17.6	0.97, s	19.3	1.09, s
22	16.5	0.87, s	16.4	0.88, s	16.5	0.87, s	16.4	0.88, s	16.4	0.88, s	16.4	0.89, s
23	21.5	1.32, s	20.1	1.21, s	21.4	1.31, s	19.9	1.22, s	19.9	1.22, s	17.0	1.11, s
24	171.9		171.0		170.2		171.0		170.9		170.1	
25	170.4		170.6		170.3		170.5		170.5		170.5	
12-CH_3_*C*O	172.1		172.1		171.8		172.0		171.9		171.9	
12-*C*H_3_CO	21.1	1.94, s	21.1	1.94, s	21.1	1.92, s	21.1	1.94, s	21.1	1.96, s	21.2	1.99, s
16-O*C*H_3_					58.0	3.52, s	57.9	3.46, s	57.9	3.46, s		
1′	34.5	3.69, m	34.5	3.70, t (7.0)	34.6	3.70, td (6.9, 1.7)	34.6	3.70, td (6.9, 2.1)	34.6	3.72, t (6.7)	34.6	3.72, t (6.9)
2′	33.5	2.56, td (6.9, 4.8)	33.5	2.56, td (6.9, 1.5)	33.4	2.56, td (6.9, 3.1)	33.5	2.56, td (7.0, 5.0)	33.8	2.56, td (6.7, 5.1)	33.5	2.57, td (6.9, 3.1)
3′	174.4		174.4		174.4		174.3		173.0		174.5	
4′									52.3	3.63, s		

^a^ Data obtained in CD_3_OD. The assignments were aided by the COSY, NOESY, HSQC, and HMBC spectra.

**Table 2 marinedrugs-20-00726-t002:** The ^13^C (150 MHz) and ^1^H (600 MHz) NMR data for **7**−**12** ^a^.

Position	7		8		9		10		11		12	
	*δ*c	*δ*_H_ (*J* in Hz)	*δ*c	*δ*_H_ (*J* in Hz)	*δ*c	*δ*_H_ (*J* in Hz)	*δ*c	*δ*_H_ (*J* in Hz)	*δ*c	*δ*_H_ (*J* in Hz)	*δ*c	*δ*_H_ (*J* in Hz)
1	40.9	1.60, m 0.64, m	40.9	1.60, m 0.64, m	40.9	1.61, m 0.65, m	40.9	1.59, m 0.64, m	40.9	1.60, m 0.68, m	40.9	1.60, m 0.68, m
2	19.5	1.64, m 1.40, m	19.5	1.64, m 1.40, m	19.5	1.66, m 1.42, m	19.5	1.65, m 1.41, m	19.6	1.66, m 1.42, m	19.5	1.66, m 1.42, m
3	43.2	1.38, m 1.16, m	43.2	1.38, m 1.15, m	43.2	1.40, m 1.18, m	43.2	1.38, m 1.16, td (13.2, 4.0)	43.2	1.39, dt (13.4, 3.8) 1.18, td (13.4, 4.1)	43.2	1.39, m 1.18, td (13.2, 3.9)
4	34.2		34.2		34.2		34.2		34.3		34.3	
5	58.1	0.87, m	58.0	0.88, m	58.0	0.89, m	58.0	0.88, m	58.1	0.92, m	58.1	0.91, m
6	19.2	1.60, m 1.49, qd (13.0, 3.3)	19.2	1.62, m 1.49, qd (13.0, 3.3)	19.2	1.64, m 1.51, m	19.2	1.63, m 1.49, m	18.9	1.64, m 1.52, qd (12.8, 3.3)	18.9	1.64, m 1.53, m
7	42.4	1.88, m 1.03, td (12.8, 3.9)	42.5	1.88, m 1.01, td (12.9, 4.1)	42.6	1.90, m 1.03, dt (13.1, 7.0)	42.5	1.89, m 1.04, m	42.0	2.02, td (12.6, 3.4) 1.03, m	41.9	2.02, m 1.03, m
8	38.3		38.3		38.5		38.6		38.5		38.5	
9	54.4	1.23, m	54.4	1.23, m	54.3	1.23, dd (13.3, 2.3)	54.3	1.28, m	53.5	1.35, dd (13.1, 2.4)	53.5	1.34, m
10	38.1		38.1		38.1		38.1		38.1		38.1	
11	21.9	1.98, m 1.75 ddd (15.2, 13.2, 2.4)	21.8	1.99, dt (14.8, 3.0) 1.75, ddd (15.2, 13.2, 2.5)	21.8	1.99, m 1.76, ddd (15.2, 13.2, 2.5)	22.3	1.91, m 1.79, m	22.6	1.97, m 1.77, ddd (15.2, 13.0, 2.2)	22.6	1.96, dd (14.8, 3.3) 1.77, m
12	75.9	5.56, t (2.8)	75.8	5.57, t (2.8)	75.9	5.60, t (2.8)	76.6	4.96, t (2.8)	75.1	5.05, dd (3.7, 2.1)	75.0	5.06, s
13	40.7		41.1		40.9		42.3		43.4		43.4	
14	47.5	1.79, m	47.3	1.81, dd (13.0, 7.1)	51.7	1.67, m	50.9	1.68, m	54.1	2.68, d (3.0)	54.1	2.68, m
15	22.8	2.12, d (14.4) 1.60, m	22.5	2.13, d (14.5) 1.64, m	28.9	2.14, dd (12.2, 6.6) 1.57, m	28.6	2.15, dd (12.8, 7.0) 1.53, m	130.3	6.03, dd (9.8, 2.6)	130.3	6.03, dd (9.7, 2.6)
16	71.3	4.02, dd (4.4, 1.3)	71.7	3.91, dd (4.4, 1.4)	68.6	4.44, dd (9.8, 6.6)	66.4	4.49, dd (9.8, 6.7)	118.8	6.25, dd (9.7, 3.3)	118.8	6.24, dd (9.7, 3.3)
17	151.4		148.8		153.9		133.4		130.3		130.3	
18	142.8		145.6		140.1		162.3		159.8		159.8	
19	33.8	0.88, s	33.8	0.88, s	33.7	0.88, s	33.7	0.88, s	33.8	0.88, s	33.8	0.88, s
20	21.8	0.85, s	21.8	0.85, s	21.7	0.86, s	21.7	0.85, s	21.8	0.86, s	21.8	0.86, s
21	17.5	0.96, s	17.5	0.97, s	17.6	0.98, s	17.7	0.99, s	19.2	1.10, s	19.2	1.10, s
22	16.5	0.88, s	16.5	0.88, s	16.4	0.88, s	16.6	0.88, s	16.4	0.89, s	16.4	0.89, s
23	19.9	1.16, s	20.1	1.18, s	21.6	1.26, s	21.6	1.26, s	17.5	1.06, s	17.5	1.06, s
24	81.7	5.39, d (1.9)	87.0	5.44, m	52.0	4.07, d (19.7) 3.97, d (19.8)	172.1		171.6		173.9	
25	170.0		170.3		171.7		50.4	4.06, d (19.6) 3.77, d (19.6)	50.5	4.06, m	50.4	4.09, m 4.01, m
12-CH_3_*C*O	172.2		172.1		172.3		172.5		172.0		172.4	
12-*C*H_3_CO	21.1	1.90, s	21.1	1.90, s	21.2	1.90, s	21.2	2.01, s	21.3	2.11, s	21.3	2.13, s
16-O*C*H_3_	57.4	3.43, s	57.4	3.43, s								
24-O*C*H_3_			50.5	3.04, s								
1′	36.2	3.68, m 3.53, dq (14.3, 7.2)	36.6	3.69, m 3.40, m	39.4	3.75, dt (13.4, 6.4) 3.58, dt (14.0, 6.7)	39.3	3.74, m 3.58, dt (13.7, 6.4)	39.5	3.73, dt (14.1, 6.3) 3.65, m	39.4	3.74, m 3.66, m
2′	34.2	2.63, dt (16.4, 7.3) 2.53, dt (16.4, 6.4)	33.5	2.64, ddd (16.4, 8.0, 6.4) 2.52, dt (16.4, 6.1)	34.1	2.60, dt (13.5, 6.6)	34.0	2.59, m	34.1	2.61, td (6.8, 6.3, 5.1)	33.9	2.64, m
3′	175.0		174.9		175.3		175.1		175.3		174.4	
4′											52.3	3.65, s

^a^ Data obtained in CD_3_OD. The assignments were aided by the COSY, NOESY, HSQC, and HMBC spectra.

**Table 3 marinedrugs-20-00726-t003:** MIC values and cytotoxicity of scalimides A–L (**1**–**12**).

Compounds	MIC (μg/mL)						MCF7 ^b^
	Gram (+) Bacteria ^a^			Gram (−) Bacteria ^a^			
	A	B	C	D	E	F	EC_50_ (μM)
**1**	32	64	16	>128	>128	>128	25.6
**2**	>128	>128	128	>128	>128	>128	73.9
**3**	32	64	8	>128	>128	>128	13.7
**4**	32	128	4	>128	>128	>128	16.0
**5**	>128	>128	>128	>128	>128	>128	26.5
**6**	64	128	32	>128	>128	>128	38.7
**7**	32	128	32	>128	>128	>128	31.5
**8**	32	128	8	>128	>128	>128	20.7
**9**	>128	>128	>128	>128	>128	>128	42.7
**10**	8	16	4	32	64	64	23.3
**11**	16	32	4	>128	>128	>128	20.6
**12**	>128	>128	>128	>128	>128	>128	42.9
Kanamycin	2	0.06	0.5	2	1	2	
Staurosporine							0.13

^a^ A: *Micrococcus luteus* (KCTC-1915); B: *Staphylococcus aureus* (KCTC-1927); C: *Bacillus subtilis* (KCTC-1021); D: *Salmonella typhimurium* (KCTC-2515); E: *Klebsiella pneumoniae* (KCTC-2690); F: *Escherichia coli* (KCTC-2441); ^b^ MCF7: breast cancer cell line.

## Data Availability

All data presented in this report are available with permission from the corresponding author upon request.

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
