# Peer review of "Isolation of Scalimides A–L: β-Alanine-Bearing Scalarane Analogs from the Marine Sponge Spongia sp."

_marinedrugs, 2022, doi:10.3390/md20110726_

Round 1
Reviewer 1 Report
A paper entitled “Isolation of Scalimides A-L: β-Alanine-bearing Scalarane Alkaloids from the Marine Sponge Spongia sp.” is submitted to Marine Drugs for further reviewing and publication. In this interesting research work, 12 new scalarane-related alkaloids were isolated and reported. Their structures were determined by spectroscopic methods. Several of these compounds were found to display antimiccrobal activity. I recommended that this research work is acceptable for publication after revisions.
1. Keywords, scalarane sesterterpenes→scalarane sesterterpene; scalimides→scalimide; the phrase “marine natural products” is not good to be list as a key word.
2. In Figure 2, compounds 6 and 11, R1 = H, R2 = H, ∆15,16→R1 = R2 = H, ∆15.
3. In Figure 2, compound 12, ∆15,16→∆15.
4. Page 3, line 83, oxygenated-methines→oxymethines. Check it throughout the text.
5. Page 3, line 92, 12-acetyl group→12-acetoxy group.
6. If it possible, please redraw Figures 3 and 5, it is very difficult to understand it.
7. In Figure 3, the COSY correlations between H2-2/H2-3 in 1; H2-1/H2-2 in 3; and H2-2/H2-3 in 5; were not found. Why?
8. Page 6, line 163, ∆15,16→∆15.
9. In Table 2, the chemical shifts, coupling patterns and coupling constants for H2-3 in compound 11 were assigned as δH 1.39 dt (16.1, 3.1) and 1.18 td (13.4, 4.4). Why? Which one is the vicinal coupling constant?
10. In Table 2, the chemical shift, coupling patterns and coupling constants for H-1' in compound 7 were assigned as δH 3.53 dq (14.3, 7.2). Why?
11. The 1H and 13C NMR data for all isolates were shown in Tables 1 and 2. However, in page 12, the 1H and 13C data for compounds 1-12 were shown in Tables S1….S12. Why?
12. Please move all the data shown in pages 12 and 13 for compounds 1-12 to the end of 3.3 Extraction and Isolation.
Too many typing and grammatical errors still found. Please check it carefully throughout the text.
Reviewer 2 Report
The article titled “Isolation of Scalimides A-L: b-Alanine-bearing Scalarane Alkaloids from the Marine Sponge Spongia sp” by Shin and colleagues presents the structures of twelve new marine natural products that possess rare modifications on a well known scalarane-type scaffold.
The manuscript is well written, with only very few errors.
In the structure elucidation argument of all of the compounds, consider using “indices of hydrogen deficiency” terminology, rather than "degrees of unsaturation".
While the authors do an excellent job at elucidating the relative configuration of the isolated compounds, there is no mention of the absolute configuration. This is a well known class of compounds and the authors should modify the text to include the elucidation of the absolute configurations of the new natural products.
Minor comments:
Page 2, line 62 insert space between “mild” and “antimicrobial”.
Page 2, line 65 insert “word” report between “and” and “their”
